# Synergetic Effects of Zinc, Boron, Silicon, and Zeolite Nanoparticles on Confer Tolerance in Potato Plants Subjected to Salinity

**Abdel Wahab M. Mahmoud [1],\*, Emad A. Abdeldaym [2]** 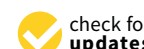 **, Suzy M. Abdelaziz [3],**
**Mohamed B. I. El-Sawy [4] and Shady A. Mottaleb [1]**

[1] Plant Physiology Division, Department of Agricultural Botany, Faculty of Agriculture, Cairo University, P.O. Box 12613 Giza, Egypt; shady.osama@agr.cu.edu.eg

[2] Vegetable Crops Department, Faculty of Agriculture, Cairo University, P.O. Box 12613 Giza, Egypt; emad.abdeldaym@agr.cu.edu.eg

[3] Department of Cross-pollinated vegetable crops, Horticulture Research Institute, P.O. Box 12611 Giza, Egypt; suzy_kamel2007@yahoo.com

[4] Department of Horticulture, Faculty of Agriculture, Kafrelsheikh University, P.O. Box 33516 Kafrelsheikh, Egypt; drmohamedelsawy@yahoo.com

\* Correspondence: mohamed.mahmoud@agr.cu.edu.eg; Tel.: +20-1009525004

**Abstract:** Salinity stress is a severe environmental stress that affects plant growth and productivity of potato, a strategic crop moderately sensitive to saline soils. Limited studies are available on the use of combined nano-micronutrients to ameliorate salinity stress in potato plants (*Solanum tuberosum* L.). Two open field experiments were conducted in salt-affected sandy soil to investigate plant growth, physiology, and yield of potato in response to soil salinity stress under single or combined application of Zn, B, Si, and Zeolite nanoparticles. It was hypothesized that soil application of nanoparticles enhanced plant growth and yield by alleviating the adverse impact of soil salinity. In general, all the nano-treatments applications significantly increased plant height, shoot dry weight, number of stems per plant, leaf relative water content, leaf photosynthetic rate, leaf stomatal conductance, chlorophyll content, and tuber yield, as compared to the untreated control. Furthermore, soil application of these treatments increased the concentration of nutrients (N, P, K, Ca, Zn, and B) in plant tissues, leaf proline, and leaf gibberellic acid hormone (GA3) in addition to contents of protein, carbohydrates, and antioxidant enzymes (polyphenol oxidase (PPO) and peroxidase (POD)) in tubers. Compared to other treatments, the combined application of nanoparticles showed the highest plant growth, physiological parameters, endogenous elements (N, P, K, Ca, Zn, and B) and the lowest concentration of leaf abscisic acid (ABA) and transpiration rate. The present findings suggest that soil addition of the aforementioned nanoparticles can be a promising approach to improving crop productivity in salt-affected soils.

**Keywords:** potato crop; salinity; nano-particles; antioxidants; hormones; leaf gas exchange; yield

## 1. Introduction

Salinity stress is one of the most severe environmental limitations to plant growth and productivity, particularly, in arid and semi-arid regions worldwide [1]. Over 6% of the world land is affected by salinity, which accounts for more than 800 million hectares [2]. Plant growth is seriously affected by salt stress, and plants adapt to this abiotic stress, in order to survive, by adopting several strategies [3,4]. Excess of salt ions in either water or soil causes significant changes in morphological, physiological and biochemical attributes of plants. In saline environments, plants take up an excessive amount of sodium

(Na$^+$) at the expense of potassium and calcium. As Na$^+$ contents of leaves, stems, and other parts increase with the increasing salinity, this might lead to nutritional imbalance that causes decreased plant growth and dry matter production. [5]. Notably, this stress also induces osmotic stress in plants, reduction in photosynthetic rate, breakdown of pigments, and imbalance of water absorption and nutrient uptake. In addition, the exposure of crops to salinity stress increases the accumulation of Na$^+$ and Cl$^-$ ions which produces reactive oxygen species (ROS), causing oxidative damage to cell components and impairing plant growth, [6,7]. Salt damage is dependent on numerous factors such as cultivars, growth stage, fertilizer types, and other environmental factors.

Potato (*Solanum tuberosum* L.) is an herbaceous species grown for its tuber, a fleshy stem with buds in the axils of leaf-scars. Potato is ranked fourth by the Food and Agriculture Organization of the United Nations (FAO) as the most important food crop [8,9]. Potato is classified as moderately sensitive to salinity, tolerating Ec$_e$ values up to 1.7 dSm$^{-1}$ with 12% yield decrease with each increase in dSm$^{-1}$ [10]. It is considered to be more sensitive during the period of initiation of the tuber bud [11]. Generally, it has been documented that potato yield significantly decreased with increasing salt concentration. For example, potato production was shown to decrease as salt concentration increases with either drip irrigation water of 1–2 dSm$^{-1}$ or surface irrigation water of 3–4 dSm$^{-1}$ [12–14]. Responses of potato crop to salinity are commonly evaluated in terms of survival ratio, vegetative growth, tuber weight, number of tubers per plant, tuber size or tuber yield [14].

In general, several approaches have been applied to minimize salinity stress for several vegetable crops such as grafting [15], inoculation with beneficial microorganisms [5,16], pre-sowing treatments [17], utilization of organic amendments [18,19] and nanoparticles [20]. It was observed that, application of nano-elements improves morphological and physiological traits of plants under normal [21] and stress conditions [22,23]. The effects of nanomaterials in plants depend on their physiochemical properties and cultivar species [24]. Further studies on vegetable crops under salinity condition reported that implementation of nanoparticles improves seed germination and antioxidant activity of crops. Chemically-modified natural or synthetic zeolites have been exploited to reduce Na+ in saline soil. The systems governing the salt removal by zeolites are primarily ion exchange, adsorption, and salt storage. Factors such as zeolite geochemical properties, pH, co-existing anions, concentration, valency, surface charge, surface area, and soil types control the ion exchange process. However, little if any literature is available on the mechanism of action of nano-zeolite in salinity control in the open field [25]. Silicon is the second most abundant mineral element in the soil and its ability to ameliorate the negative effect of NaCl on plant growth rate has been widely reported. Several studies reported that salt stressed plants treated with silicon have shown salt tolerance. Recently, various sources of silica have been used as fertilizers, or applied for promoting the growth of crops while their side effects are still unclear [26–28]. On the other hand, zinc is well-known as a vital activator of several enzymes in plants. Besides, it is directly involved in the biosynthesis of growth substances such as auxin, which produces more plant cells, thus increasing the dry matter. Hussein et al. [29] reported that, foliar application of ascorbic acid in combination with zinc sulfate increased the plant height and total plant biomass under salinity conditions. Also, Hussein et al. [30] cited that, the foliar application of nano-Zn led to mitigating the adverse effect of salinity and confirmed that diluted seawater could be used in the irrigation of cotton plant. Hereabout, Martínez-Ballesta et al. [31] documented that, the activity of specific membrane components can directly be influenced by boron under saline condition through regulating the water uptake and water transport via the functions of certain aquaporin isoforms.

Therefore, the objective of the present work was to examine and explain the influence of soil amendment using zeolite with silicon (Si), zinc (Zn), or boron (B) in the form of nanoparticles, individually or combined, on some morphological, physiological, biochemical and photosynthetic characteristics of potato plants under field salinity conditions.

## 2. Materials and Methods

### 2.1. Experimental Location

The present research was carried out in an open field at a newly reclaimed desert area with salinity conditions (>3 ds/m) located in Wadi El-Notron, Beheira Governorate, Egypt (Longitude: 28° 54′ E, Latitude: 28° 20′ N and Altitude: 125 m). Before cultivation, physical and chemical analyses of the soil were performed (Table 1) at the Soil, Water and Environment Research Institute (SWERI), Agriculture Research Center (A.R.C) according to [32,33]. Before planting, chemical characteristics of the compost, obtained from SWERI, were carried out (Table 2) as described by [34].

**Table 1.** Analysis of experimental site before fertilizer application.

| Parameters | Soil Depth (cm) | |
|---|---|---|
| | 0–30 | 30–60 |
| **Particle-size distribution [%]** | | |
| Sand | 90.10 | 90.00 |
| Silt | 6.90 | 6.50 |
| Clay | 3.00 | 3.50 |
| Textural class | Sand | Sand |
| Saturation water content [$cm^3\ cm^{-3}$] | 0.385 | 0.396 |
| Field capacity [$cm^3\ cm^{-3}$] | 0.213 | 0.218 |
| Permanent wilting point [$cm^3\ cm^{-3}$] | 0.057 | 0.057 |
| Available water [$cm^3.cm^{-3}$] | 0.156 | 0.161 |
| Bulk density [$mg\ m^{-3}$] | 1.64 | 1.65 |
| Saturated hydraulic conductivity, [$cm\ day^{-1}$] | 240.00 | 234.00 |
| Organic matter [%] | 0.31 | 0.25 |
| Calcium carbonates [%] | 4.80 | 3.71 |
| pH (1:1, soil: water suspension) | 7.70 | 7.81 |
| EC (1: 1, soil: water extract) [$dS.m^{-1}$] | 4.02 | 4.13 |
| **Soluble Cations, [Cmole(+) $Kg^{-1}$ soil]** | | |
| $Ca^{2+}$ | 13.85 | 13.41 |
| $Mg^{2+}$ | 12.15 | 10.59 |
| $Na^+$ | 8.10 | 10.25 |
| $K^+$ | 6.00 | 6.05 |
| **Soluble Anions, [Cmole(–) $Kg^{-1}$ soil]** | | |
| $CO_3^{2-}$ | - | - |
| $HCO_3^-$ | 11.92 | 9.75 |
| $Cl^-$ | 14.00 | 10.50 |
| $SO_4^{2-}$ | 15.08 | 21.30 |
| **Available nutrients [mg $Kg^{-1}$ soil]** | | |
| N | 16.21 | 13.12 |
| P | 7.78 | 6.21 |
| K | 46.50 | 45.89 |
| Fe | 9.20 | 12.00 |
| Mn | 1.63 | 1.50 |
| Cu | 2.10 | 1.15 |
| Zn | 2.00 | 1.61 |
| **B** | 0.23 | 0.21 |

**Table 2.** Chemical properties of applied compost.

| Property | Value |
|---|---|
| Moisture content [%] | 25 |
| pH [1:5] | 7.5 |
| EC (1: 5 extract) [$dS\ m^{-1}$] | 3.1 |
| Organic-C [%] | 33.11 |
| Organic matter [%] | 70 |
| Total-N [%] | 1.82 |
| Total-K [%] | 1.25 |
| C/N ratio | 14:1 |
| Total-P [%] | 1.29 |
| Fe [ppm] | 1019 |
| Mn [ppm] | 111 |
| Cu [ppm] | 180 |
| Zn [ppm] | 280 |
| Total content of Bacteria [$CFU \cdot g^{-1}$] | $2.5 \times 10^7$ |
| Phosphate dissolving Bacteria [$CFU \cdot g^{-1}$] | $2.5 \times 10^6$ |
| Weed seeds | 0 |

### 2.2. Plant Material and Harvest Dates

The experiments were repeated for two successive seasons (2017/2018 and 2018/2019). As to the first season, planting was done on 9 October 2017 while harvesting was carried out on 10 February 2018. As to the second season, planting was done on 12 October 2018 while harvesting was carried out on 14 February 2019. Imported potato tuber seeds (Solanum tuberosum, L., Diamont cultivar) were planted with 40 cm between seeds, 70 cm between rows and at 12−15 cm depth.

### 2.3. Soil Preparation

Before planting, the soil was first mechanically ploughed deeply (35–45 cm) and planked twice till the soil surface had settled.

### 2.4. Organic Matter (Compost)

Compost (plant residues from legumes corps), at the rate of 2.1 t ha$^{-1}$ was incorporated into the soil 12 days before planting.

### 2.5. Irrigation System

Irrigation water was supplied through a drip irrigation network using (4.0 L h$^{-1}$) drippers [35].

### 2.6. Chemical Fertilizers

As recommended by the Ministry of Agriculture and Land Reclamation (MALR, Egypt), chemical fertilizers were added at the rate of 43.3 kg ha$^{-1}$ mono-superphosphate (15.5% P), 21.6 kg ha$^{-1}$ potassium sulphate (48% K) and 43.3 kg ha$^{-1}$ ammonium sulphate (20.5% N) during soil preparation, and immediately after planting 43.3 kg ha$^{-1}$ mono-superphosphate, 21.6 kg ha$^{-1}$ potassium sulphate, and 32.4 kg ha$^{-1}$ ammonium sulphate.

### 2.7. Nano-Zeolite

Nano-zeolite was prepared according to [36] then was loaded with nitrogen (Figure 1A) and (Table 3) according to [37]. Transmission electronic microscope examination and imaging (TEM) were done at the Research Park of Faculty of Agriculture, Cairo University (FA-CURP). Nano-zeolite was added at a rate of 1.3 L ha$^{-1}$ through the irrigation network 15 days before planting and 20, 35, 45, and 70 days after planting.

**Table 3.** Chemical composition of nano-zeolite loaded nitrogen.

| Chemical Composition (%) | SiO$_2$ | TiO$_2$ | Al$_2$O$_3$ | Fe$_2$O$_3$ | FeO | MnO | MgO | CaO | Na$_2$O | K$_2$O | SrO | P$_2$O$_3$ | N |
|---|---|---|---|---|---|---|---|---|---|---|---|---|---|
| | 45.50 | 2.81 | 13.30 | 5.40 | 8.31 | 0.51 | 6.30 | 9.52 | 2.83 | 0.87 | 0.22 | 0.67 | 2.70 |
| Trace Elements (ppm) | Ba | Co | Cr | Se | Cu | Zn | Zr | Nb | Ni | Rb | Y | | |
| | 10 | 1.2 | 35 | 0.8 | 19 | 64 | 257 | 13 | 55 | 15 | 22 | | |

### 2.8. Nano Zinc, Boron, and Silicon

All the used reagents were of analytical grade and the nanoparticles were prepared from their precursors. Zinc in the form of zinc chloride (ZnCl$_2$), boric anhydride in the form of boric oxide, (B$_2$O$_3$) and silicon in the form of silicon tetrachloride (SiCl$_4$) were purchased from Sigma Chemical Co. (St. Louis, MO, USA).

Nanoparticles were obtained by the top to bottom molecular chemical method [38]. Nano zinc (Figure 1B) was prepared from an aqueous solution of zinc chloride. Sodium hydroxide solution was added slowly in a molar ratio of 1:2 under vigorous stirring for 8 h. The obtained precipitate was filtered and washed thoroughly with deionized water in a mixed water/toluene system using a high-speed stirrer, and then washed again with ionized water alone for 3 h. The precipitate was dried

in an oven at 100 °C, (2.75 g nano-particles powder) then exposed to 1.5 psi of pressure for 3 days discontinuously (7 h per day) [39].

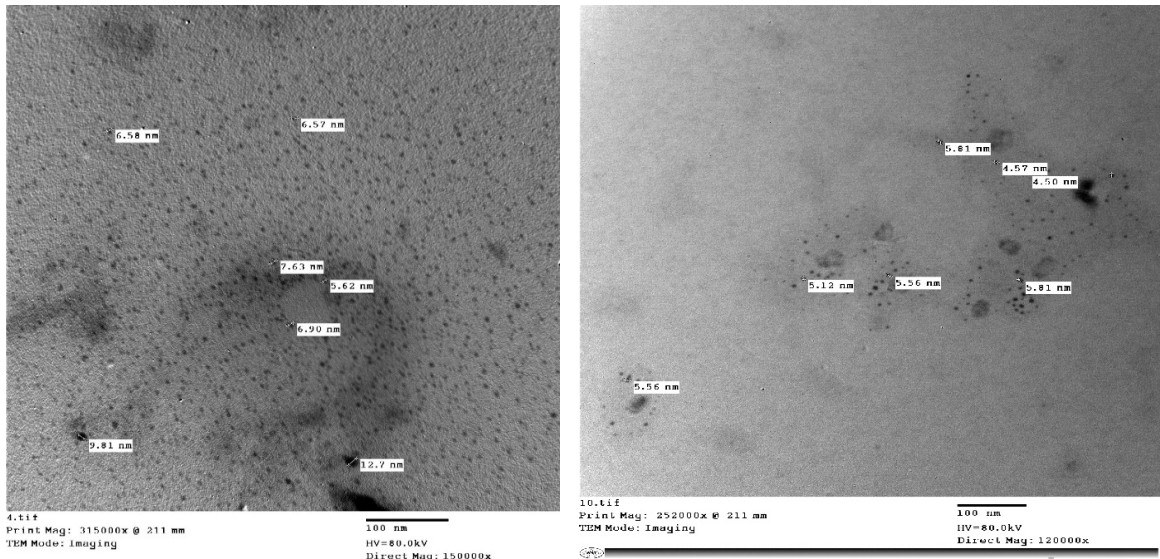

Image (A) Nano zeolite                    Image (B) Zinc nano-particles

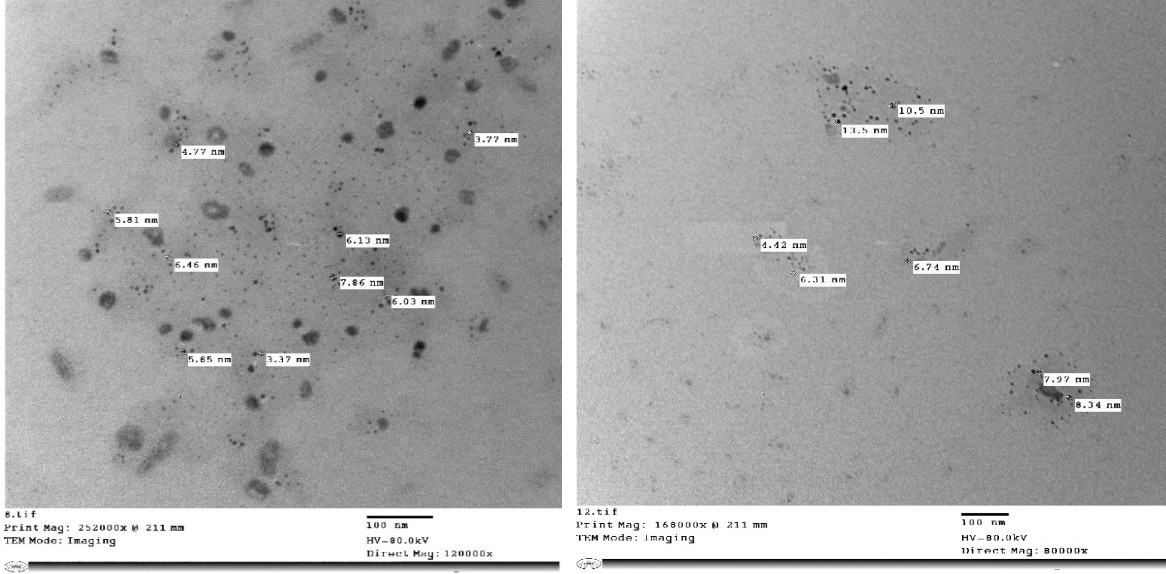

Image (C) Boron nano-particles           Image (D) Silicon nano-particles

**Figure 1.** Image of Nano zeolite (**A**), Zinc nano-particles (**B**), Boron nano-particles (**C**) and Silicon nano-particles (**D**).

For nano-boron (Figure 1C), $B_2O_3$ and Mg were mixed in 2:1 (*w:w*) ratio with ball-to-powder weight ratio (32:1). Initial materials containing stoichiometric amounts of reactants were prepared in an exothermic reaction. The milling container was a hardened steel vial with a capacity of 250 cm$^3$ and the milling media was hardened steel balls. The container was filled with argon (purity: 99%) to hinder oxidation of products and the powders milled during the reaction. With a rotation speed of vial of 440 rpm, milling was carried out for 10 h. Milled powders were leached by HCl solution (Merck, 28%) at a temperature between 60–70 °C by a heater-magnetic stirrer to remove other solid by-products such as MgO. The final product was obtained (3.2 g nano-particles powder) after centrifuging at 500 rpm for 30 min, decanting, washing (deionized water, 4 h, 100 °C). Drying treatment was carried out in an oven at 50 °C for 12 h [39].

As for nano-silicon (Figure 1D), mild reagents (3-aminopropyl) tri-methoxysilane and ascorbate sodium were used through a quick reaction in a commonly used round bottom flask at room temperature and pressure. Trimethoxysilane (97%) and ascorbate sodium were prepared (1:4 m ratio) in aqueous solution while stirring. Then, 1.25 mL of 0.1 M ascorbate sodium was added to the above mixture by stirring for 40 min, and then exposed to 1.5 psi of pressure for 5 days discontinuously (8 h per day). The precipitate (2.4 g nano-particles powder) was dried in an oven at 80 °C for 10 h. [40].

Zinc, boron. and silicon nano-particles were added to the soil individually and in combination at concentrations of 20, 12, and 15 ppm, respectively, 25, 40, 50, and 75 days after planting.

During the two successive seasons, the treatments were as follows:

- Recommended dose of chemical fertilizers (NPK) as control T0
- Nano zeolite　T1
- Nano zinc　T2
- Nano boron　T3
- Nano silicon　T4
- Nano zinc, boron and silicon　T5
- Nano zeolite, zinc, boron and silicon　T6

### 2.9. Data Recorded

The following data was recorded:

A. morphological traits

- Plant height [cm]
- Shoot fresh and dry weight [g plant$^{-1}$]
- Relative water content [%]
- Number of stems per plant
- Number of tubers per plant
- Photosynthetic rate [$CO_2$ m$^{-2}$ s$^{-1}$]
- Intercellular $CO_2$ concentration [ppm]
- Stomatal conductance [$H_2O$ m$^{-2}$ s$^{-1}$]
- Tuber yield [t ha$^{-1}$].
- Water use efficiency [µmol mmol$^{-1}$].

B. Relative Water Content

After 15 days of treatment, samples from the top fully expanded leaves were taken for relative water content (RWC) determination. The fresh weight (FW) of five leaf discs was recorded and then the discs were immersed in deionized water for 4 h. The wet surface of the turgid leaf discs was blot-dried quickly before weighing (TW). The leaf discs were then dried for 72 h at 70 °C in an oven, and the dry weight (DW) was then measured. The RWC was calculated and expressed in percentage based on the formula: RWC = (FW − DW/TW − DW) × 100

C. Net Photosynthesis

Measurements of net photosynthesis on an area basis [µmol $CO_2$ m$^{-2}$ s$^{-1}$], leaf stomatal conductance (mol $H_2O$ m$^{-2}$ s$^{-1}$), intercellular $CO_2$ concentration (ppm) and water use efficiency of five different leaves per treatment were monitored using a LICOR 6400 (Lincoln, NE, USA) infrared gas analyzer (IRGA). Light intensity (photosynthetically active radiation, PAR) within the sampling chamber was set at 1500 (µmol m$^{-2}$ s$^{-1}$), using a Li-6400-02B LED light source (LI-COR). The $CO_2$ flow into the chamber was maintained at a concentration of 400 µmol mol$^{-1}$ using a LI-6400-01 $CO_2$ mixer (LI-COR).

D. Chemical Analysis

The plant material was dried in an electric oven at 70 °C for 24 h according to [41], then finely ground for chemical determination of elements. The wet digestion of 0.2 g plant material with sulfuric and perchloric acids was carried out on samples by adding concentrated sulfuric acid (5 mL) and the mixture was heated for 10 min. Then 0.5 mL perchloric acid was added and heating continued till a clear solution was obtained [33,41].

Total nitrogen content of the dried leaves was determined using the modified micro-Kjeldahl method as described by [41]. Phosphorus was determined colorimetrically by using the chloro-stannous molybdophosphoric blue color method in sulfuric acid according to [33]. Potassium and sodium concentrations were determined using the flame photometer apparatus (CORNING M 410, Halstead, UK).

Total carbohydrates in plant samples were determined by the phosphomolybdic acid method according to [41]. A sample of 2 g was crushed with 10 mL 80% ethanol in a mortar and pestle, then filtered through a Whatman filter paper. The filter and residue were collected separately. The alcohol residue was taken into a 250 mL conical flask. Then 150 mL distilled water and 5 mL concentrated HCl were added into the flask. The residue was hydrolyzed for 30 min and cooled to room temperature. $Na_2CO_3$ was then added slowly until the extract became neutral (pH = 7). The extract was filtered, and the residue discarded. The filtrate was taken into a conical flask and condensed in a water bath for up to 3–4 min. Distilled water was added to the filtrate, then filtered, after mixing. The residue was discarded and the volume of filtrate served for reducing sugar. Then 20 mL of this filtrate was taken into a 150 mL conical flask and 2 mL of concentrated HCl was added to it. It was then hydrolyzed for 30 min and cooled to room temperature. $Na_2CO_3$ was slowly added until the extract became neutral (pH = 7). This extract was filtered and the residue was discarded and the final volume of the filtrate was measured and used as a sample for total sugar. A sample aliquot of 0.5 mL was taken into a test tube and 1 mL of Somogy's reagent was added. The test tubes were placed in boiling water bath for 30 min, cooled to room temperature, and 1 mL of arsenomolybdate reagent was added. The content was mixed and diluted to a volume of 10 mL and its absorbance was spectrophotometrically measured at 560 nm. Concentrations of Ca, Zn, and B in plant samples were determined using atomic absorption spectrophotometer with air-acetylene and fuel (PyeUnicam, model SP−1900, Cambridge, UK).

Plant pigments: Total chlorophyll contents were measured by spectrophotometer and calculated according to the equation described by Moran et al. [42].

*2.10. Endogenous Phytohormones*

Determinations of gibberellic acid (GA) and abscisic acid (ABA) were performed according to [43]. The quantification of the endogenous phyto-hormones was carried out with Ati-Unicum gas–liquid chromatography, 610 Series, equipped with flame ionization detector according to the method described by [44] where freeze-dried plant samples (equivalent 6 g FW) were ground to a fine powder using a mortar and pestle. The powdered material was extracted three times (once for 3 h and twice for 1 h) with methanol (80% *v/v*, 15 mL/g FW), supplemented with butylated hydroxy toluene (2,6–di-tert-butyl-p-cresol) as an antioxidant, at 4 °C in darkness. The extract was centrifuged at 4000 rpm. The supernatant was transferred into flasks wrapped with aluminum foil and the residue was twice extracted again. The supernatants were combined and the volume was reduced to 10 mL at 35 °C under vacuum. The aqueous extract was adjusted to pH 8.6 and extracted three times with an equal volume of pure ethyl acetate. The combined alkaline ethyl acetate extract was dehydrated over anhydrous sodium sulphate then filtered. The filtrate was evaporated to dryness under vacuum at 35 °C and re-dissolved in 1 mL absolute methanol.

*2.11. Free Proline*

Free proline content was extracted from the leaf tissues according to the method described by [45]. Using a cold extraction procedure by mixing 20–50 mg fresh weight aliquots with 0.5−1 mL of ethanol:

water (60:80 *v/v*). The resulting mixture was left overnight at 5 °C and then centrifuged at 15,000 *g* (4 min). Then 1 mL of alcoholic extract was diluted with 10 mL of distilled water and 5 mL of ninhydrin (0.125 g ninhydrin, 2 mL of 6 mM $NH_3PO_4$, 3 mL of glacial acetic acid), and 5 mL of glacial acetic acid were added, and the mixture was placed in a boiling water bath for 45 min at 100 °C. The reaction was stopped by placing the test tubes in cold water. The cold extraction procedure was repeated on the pellet, and then the pooled supernatants were used for the analysis using a PD−303 model spectrophotometer.

### 2.12. Starch Content (%)

Starch content was determined in the tuber's dry matter using the method described by [41] with some modifications. Milled samples were added (~80 mg) to tubes, then 0.2 mL of aqueous ethanol (80% *v/v*) was added to the sample to aid dispersion. The tube was stirred on a vortex mixer and then 3 mL of thermostable a-amylase was added in buffer (pH 7.0) and then incubated in a boiling water bath for 6 min, vortexing vigorously after 4 and 6 min to ensure complete homogeneity. The tube was then placed in a bath at 50 °C, and 4 mL of 200 mM sodium acetate buffer (pH 4.5) added. The tube was then centrifuged at 13,000 rpm for 4 min and 0.1 mL aliquot of each sample transferred to the bottom of a 16 × 120 mm glass test tube and incubated at 50 °C for 20 min. Finally, the absorbance was measured spectrophotometrically at 630 nm against the reagent blank.

### 2.13. Peroxidase (POD)

The peroxidase amount was estimated using the method proposed by [46,47] where (0.5 g) was frozen in liquid nitrogen to extract the peroxidase enzyme. The samples were ground with 10 mL extraction buffer (50 mM phosphate buffer, pH 7, containing 0.5 mM EDTA and 2% PVPP (*w/v*) and centrifuged at 3930 rpm for 20 min. The resultant supernatant was used to determine the peroxidase activity using a spectrophotometric method by the formation of guaiacol in l mL reaction mixture (450 μl 25 mM guaiacol, 450 μl 225 mM $H_2O_2$) and 100 μl crude enzymes.

### 2.14. Polyphenol Oxidase (PPO)

The total amount of polyphenol oxidase enzyme was determined using the method described by [47]. The absorbance was measured spectrophotometrically at 495 nm and the enzyme concentration was calculated.

### 2.15. Data Analysis

The experimental design was randomized with a block design using eight replicates for each treatment. Data were subjected to statistical analysis using ANOVA at 5% significance level. The difference between treatments was then analyzed using DMRT (Duncan Multiple Range Test) at 5%.

## 3. Results and Discussion

### 3.1. Impacts of Soil Salinity on Plant Growth Parameters and Relative Water Content

Under saline conditions, exogenous addition of Zn, Si, B, and zeolite nanoparticles, single or combined, significantly improved potato growth parameters compared to control (T0) (Table 4). Similar trends were observed in the relative water content of potato leaves. Under salinity stress, the relative water content of potato leaves increased by 8.2% at the first season, and 8.4 % at the second one under the combined treatments in comparison to control. In addition, the maximum plant growth was recorded in potato plants treated by the combined elements (T6). Plant height, fresh and dry weight of potato shoots increased by 27.6%, 13.3%, and 21.3%, respectively in the first season, compared to 30.81%, 22.31%, and 22.73, respectively in the second season, compared to control. The results revealed that salinity stress had a negative impact on vegetative growth parameters and relative water content which is in accordance with the results reported by Reza and Roosta [48].

**Table 4.** Growth parameters and relative water content of potato plants grown in saline soil due to nanoparticles application.

| Treatment | Plant Height (cm) | | Shoot Fresh Weight (g) | | Shoot Dry Weight (g) | | Relative Water Content (%) | | Number of Stems Per Plant | |
|---|---|---|---|---|---|---|---|---|---|---|
| | S[1] | S[2] | S[1] | S[2] | S[1] | S[2] | S[1] | S[2] | S[1] | S[2] |
| T0 | 27.44 ± 1.78 [c] | 27.71 ± 4.13 [d] | 223.4 ± 14.5 [d] | 210.8 ± 16.3 [e] | 53.35 ± 4.95 [c] | 51.82 ± 3.3 [c] | 74.6 ± 5.73 [b] | 76.5 ± 6.32 [b] | 2.3 ± 0.65 [b] | 2.6 ± 0.36 [b] |
| T1 | 32.81 ± 5.48 [b] | 35.14 ± 2.93 [b] | 235.8 ± 12.01 [c] | 240.2 ± 13.9 [c] | 57.88 ± 5.88 [b] | 63.08 ± 6.7 [a] | 83.6 ± 4.41 [a] | 81.8 ± 7.17 [a] | 3.0 ± 0.41 [a] | 2.8 ± 0.26 [b] |
| T2 | 30.12 ± 1.85 [b] | 29.83 ± 3.53 [c] | 239.5 ± 10.31 [b] | 238.3 ± 18.1 [d] | 58.87 ± 6.04 [b] | 59.68 ± 7.3 [b] | 81.4 ± 6.8 [a] | 80.95 ± 3.66 [a] | 2.5 ± 0.22 [b] | 2.7 ± 0.25 [b] |
| T3 | 29.19 ± 3.12 [c] | 27.65 ± 2.78 [d] | 235.3 ± 19.65 [c] | 239.4 ± 20.7 [c] | 56.89 ± 4.6 [b] | 58.96 ± 3.92 [b] | 73.8 ± 7.90 [b] | 78.7 ± 9.42 [b] | 2.7 ± 0.52 [b] | 2.6 ± 0.44 [b] |
| T4 | 30.35 ± 4.1 [b] | 30.06 ± 4.45 [c] | 240.2 ± 11.23 [b] | 235.6 ± 23.5 [c] | 57.22 ± 1.99 [b] | 59.33 ± 6.07 [b] | 75.8 ± 6.45 [b] | 77.5 ± 8.41 [b] | 2.4 ± 0.38 [b] | 2.6 ± 0.34 [b] |
| T5 | 33.17 ± 5.37 [b] | 35.72 ± 2.36 [b] | 244.7 ± 21.23 [b] | 251.5 ± 13.2 [b] | 60.47 ± 2.24 [b] | 63.92 ± 4.5 [a] | 86.5 ± 7.2 [a] | 85.7 ± 5.31 [a] | 3.1 ± 0.35 [a] | 3.3 ± 0.42 [a] |
| T6 | 37.89 ± 2.82 [a] | 40.05 ± 3.91 [a] | 257.8 ± 6.31 [a] | 263.6 ± 9.31 [a] | 67.78 ± 3.4 [a] | 66.7 ± 3.10 [a] | 85.6 ± 5.21 [a] | 83.5 ± 4.50 [a] | 3.6 ± 0.43 [a] | 3.2 ± 0.56 [a] |

Values with different letters show significant differences at $p \leq 0.05$ (LSD). (T0) = control, (T1) = n-Zeolite, (T2) = n-Zn, (T3) = n-B, (T4) = n-Si, (T5) = n-Zn + Si +B, (T6) = n-zeolite+ Zn + Si + B., (n) means nanoparticles, (S) = seasons '1 and 2'.

The detrimental mechanisms that salinity stress imposes on plants include osmotic stress, toxicity due to Na cations, and nutritional disorders [49]. Salinity stress imposes an osmotic stress that starts right after the salt concentration surrounding the roots increases beyond the specific threshold level of tolerance (different for each crop), causing shoot growth rate to decrease significantly. Thus, it is of crucial importance to overcome such osmotic stress using several adaptation strategies [49]. Reduction in growth under saline conditions has been associated with inhibition of cell division and expansion [50] as well as disruption of biochemical and physiological processes of plants. In a previous study, reduced plant growth under severe salinity was shown to cause a decline in the carbon assimilation rate attributed to stomatal restriction and/or metabolic diminishing which results in growth inhibition [51].

It has been widely documented that negative effects of salt stress can be ameliorated through the application of different micronutrients to plants. For example, the addition of Si significantly improved the dry matter yields of salt stressed barley [52]. Also, Bao-Shan et al. [53] reported improved seedling growth and quality, including mean height when applying exogenous nano-$SiO_2$ to Changbai larch (*Larix olgensis*) seedlings. Furthermore, Kalteh et al. [54] observed that nano-Si might reduce the negative effects of high salinity on growth and development of basil. Moreover, silicon seems to play a role in improving water status of plants under salinity stress. For example, it was documented that adding Si caused a great decrease in cell-sap concentration of salt stressed barley, suggesting that Si might have a positive effect in increasing water-retention capacity in plants subjected to salinity [50]. Also, Romero-Aranda et al. [55] showed that when treating salt stressed tomato plants with 2.5 mM Si, their water content increased by 40% as compared to control plants, leading to an increased turgor. Moreover, the addition of Si to salt stressed wheat plants restored their relative water content (RWC) to the levels measured in control plants [56]. Regarding zinc, Raliya et al. [57] reported that nano-ZnO induced a significant improvement of biomass, shoot and root growth in *Cyamopsis tetragonoloba*. Also, Raliya et al. [58] revealed that nano-ZnO showed beneficial effects on stem height and root volume of mung bean. Likewise, a considerable amount of research has been published demonstrating benefits of zeolite in ameliorating the detrimental effects of salinity on plant growth. For instance, the application of zeolite to salt-stressed radish plants was reported to significantly increase the fresh and dry weights of shoots and roots [59]. Similar beneficial effects using zeolite amendments were observed in barley plants [60] and faba bean plants [61] subjected to salinity. Zeolite seems to act on plant growth enhancement under salinity stress in two ways. First, it significantly increases nitrogen availability to plants, consequently enhancing the synthesis of chlorophyll, antioxidant enzymes, and other structural components of the plant. Second, the water retention capability of zeolite considerably increases the relative water content of salt-stressed plants by increasing the availability of water and its absorption by roots [62]. On the other hand, our results of growth parameters together with the reports of [63,64] suggest that there is an evident synergistic effect of combining Zn, Si, B, and zeolite nanoparticles leading to improving the vigor of the plants grown under salinity conditions. For example, Panhwar et al. [65] reported that combined soil application of Zn and B significantly improved the plant height, root length, chlorophyll concentration, and the dry weight of shoot and root. These results support, once again, the beneficial role of those elements in activating the plant growth-related enzymes and reducing toxicity of sodium ($Na^+$) under salinity conditions [66–68].

### 3.2. Impacts of Soil Salinity on Leaf Chlorophyll and Photosynthetic Parameters

The experimental results showed that the total chlorophyll content and photosynthetic parameters of potato leaves were significantly affected by soil salinity (Figure 2). The content of total chlorophyll of potato plants significantly increased under treatment T6 compared to control (T0), as shown in (Figure 2A). Most notably, treatment T6 recorded the highest values of photosynthesis rate (Figure 2B), stomatal conductance (Figure 2C), intercellular $CO_2$ concentration (Figure 2D), and water-use efficiency of potato leaves (Figure 2E) and reduced the leaf transpiration rate (Figure 2F) as compared to control (T0), in both seasons. Similar results were previously reported by Ahmad et al. [69] who found significant

reduction of total chlorophyll content and photosynthetic parameters at high salinity concentrations. Net assimilation rates of $CO_2$ become inhibited under accumulation of salt in saline conditions and, consequently, restrict the $CO_2$ supply to the plants. This occurs as a consequence of stomatal closure and/or inhibition of $CO_2$ fixation [70]. Salinity also has a great detrimental effect on electron transport chain, photophosphorylation, and enzymatic activities of plants [50,71,72]. On the other hand, scientists have reported that supplementary application of Zn, Si, B, and zeolite nanoparticles enhanced the chlorophyll formation and photosynthetic activity of plants [21,73–75]. Silicon supplementation was widely reported to improve photosynthesis parameters under salinity stress conditions. Yeo et al. [76] mentioned that the growth reduction caused by salinity in rice plants can be significantly improved by addition of Si which increases the $CO_2$ assimilation rate and stomatal conductance. Similarly, Shah et al. [77] reported that the addition of Si to salt-stressed wheat plants completely restored chlorophyll content to the level of control plants. Moreover, Mateos-Naranjo et al. [78] observed significantly higher values of maximum quantum efficiency of PSII photochemistry (*Fv/Fm*), maximum fluorescence (*Fm*), and quantum efficiency of PSII (FPSII) in salt stressed halophyte *Spartina densiflora* under Si supplementation as compared to control plants. That was also the case with nano-$SiO_2$ which improves growth and development of plant through incrementing gas exchange and chlorophyll fluorescence characteristics. Such effects are accomplished by biosynthesis of photosynthetic pigments as well as by improving the activity of carbonic anhydrase [79,80]. It is worth noting that carbonic anhydrase supplies $CO_2$ to Rubisco enzyme consequently improving photosynthesis; similar results were reported in the literature regarding zinc nanoparticles. For example, nano-ZnO $Fe_3O_4$ was applied to *Moringa peregrina* under salinity conditions and was reported to increase total chlorophyll and carotenoids [81].

Also, Torabian et al. [82] used nano-ZnO on salt-stressed sunflower plants and observed an increase in the $CO_2$ assimilation rate, maximum potential quantum efficiency of photosystem II (Fv/Fm), and chlorophyll concentrations. In addition to its role in carboxylation, Zn seems also to play a critical role in stomatal closure in plant leaves [83]. Regarding boron, an antagonistic effect was reported between salinity and high boron concentrations in several horticultural crops [84–86]. This might be attributed to reduced transpiration rates caused by boron being transported through the xylem [87]. This effect, however, could also be due to the toxic effects of boron. At high salinity levels, plants may be more susceptible to toxicity from other ions [63,74,85,87]. Regarding zeolite, it was reported that its application reduces leaf senescence in salt-stressed sorrel plants through increasing leaf chlorophyll content and decreasing its degradation [88]. As mentioned above, zeolite application may also indirectly increase photosynthetic rate of salt-stressed plants by improving water and nutrient availability thanks to its high sorption capacity. Again, our results of photosynthetic parameters confirm the beneficial synergistic effect of using combined Zn, Si, B, and Zeolite nanoparticles. Similar results were reported in salt-stressed canola [89] and onion [62] plants, where combined zeolite, selenium, and silicon application led to increasing the seed yield via increasing water and nitrogen supply and consequently, improving the photosynthetic rate of amended salt-stressed plants.

### 3.3. Impacts of Soil Salinity on Endogenous Elements of Potato Plants

In both seasons, soil salinity caused a significant reduction in endogenous elements content of plant tissue ($p < 0.05$). Our findings showed that all treatments increased the endogenous concentration of N, P, K, Ca, Zn, and B elements and reduced the Na concentration in plant tissue, as compared to control (T0) (Table 5).

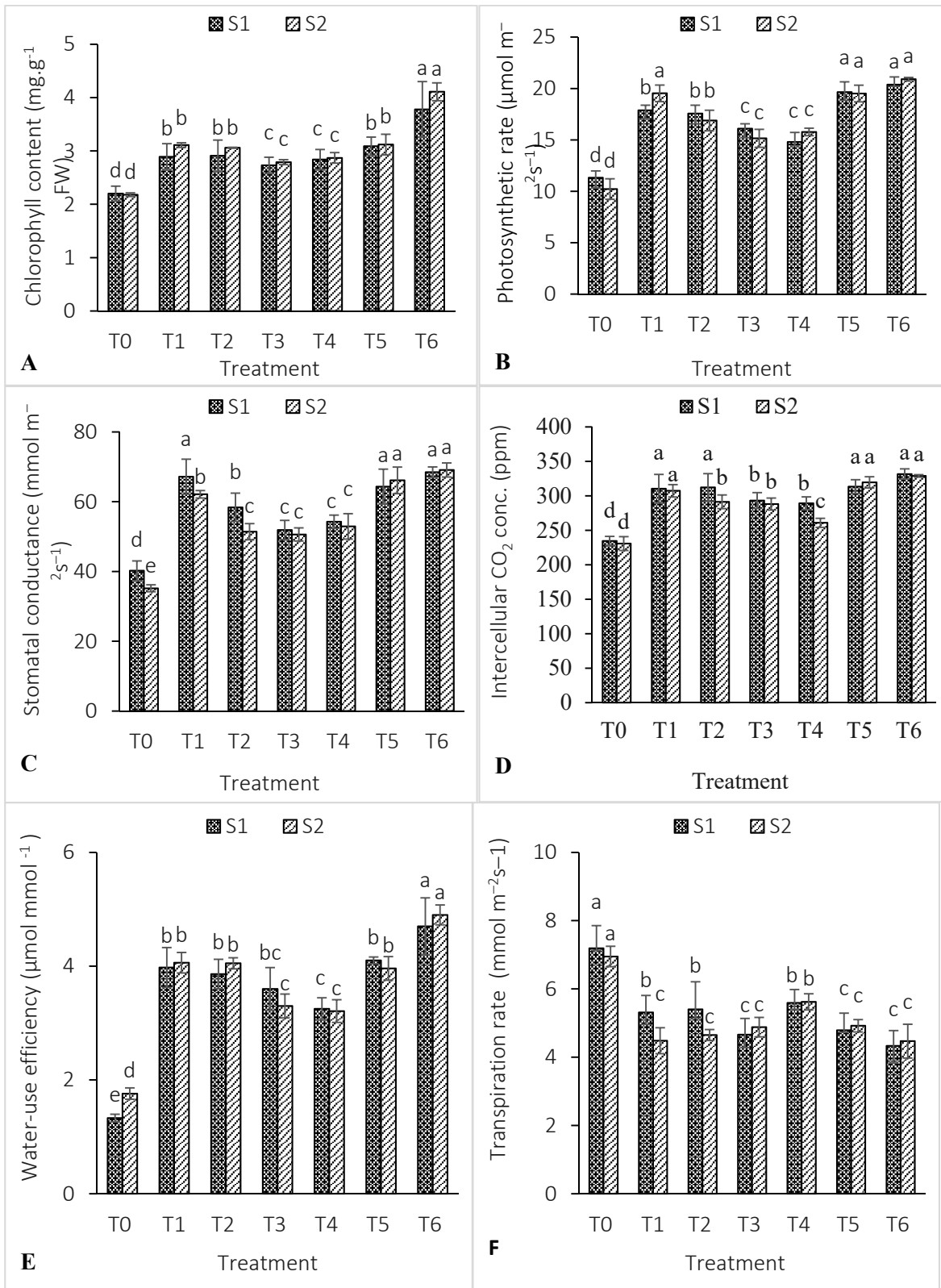

**Figure 2.** Change in (**A**) leaf chlorophyll content, (**B**) photosynthesis rate, (**C**) stomatal conductance, (**D**) intercellular $CO_2$ conc., (**E**) water use efficiency, and (**F**) transpiration rate in saline soil due to nanoparticles application. Columns with similar letters show no significant differences at $p \leq 0.05$ (LSD). Bar indicates standard deviation (SD). Treatments were organized as follow: (±SD). (T0) = control, (T1) = n-Zeolite, (T2) = n-Zn, (T3) = n-B, (T4) = n-Si, (T5) = n-Zn + Si +B, (T6) = n-zeolite + Zn + Si + B, (S) = seasons '1 and 2'.

**Table 5.** Leaf endogenous nutrient content of potato plants grown in saline soil due to nanoparticles application.

| Treatment | N (%) | | P (%) | | K (%) | |
|---|---|---|---|---|---|---|
| | S1 | S2 | S1 | S2 | S1 | S2 |
| T0 | 2.1 ± 0.24 [c] | 1.8 ± 0.21 [d] | 0.17 ± 0.048 [d] | 0.19 ± 0.057 [d] | 3.41 ± 0.34 [c] | 3.39 ± 0.18 [c] |
| T1 | 2.9 ± 0.35 [b] | 2.8 ± 0.38 [b] | 0.26 ± 0.05 [b] | 0.28 ± 0.008 [b] | 6.01 ± 0.40 [a] | 6.03 ± 0.35 [a] |
| T2 | 2.9 ± 0.36 [b] | 2.8 ± 0.37 [b] | 0.28 ± 0.019 [a] | 0.31 ± 0.065 [a] | 5.58 ± 0.26 [b] | 6.00 ± 0.42 [a] |
| T3 | 2.7 ± 0.26 [b] | 2.8 ± 0.26 [b] | 0.27 ± 0.01 [b] | 0.29 ± 0.042 [a] | 5.89 ± 0.37 [b] | 6.06 ± 0.46 [a] |
| T4 | 2.6 ± 0.42 [c] | 2.5 ± 0.51 [c] | 0.25 ± 0.055 [c] | 0.26 ± 0.07 [c] | 5.88 ± 0.31 [b] | 5.79 ± 0.31 [b] |
| T5 | 3.1 ± 0.28 [a] | 3 ± 0.25 [a] | 0.26 ± 0.07 [c] | 0.28 ± 0.009 [b] | 6.04 ± 0.47 [a] | 6.00 ± 0.52 [a] |
| T6 | 3.5 ± 0.22 [a] | 3.4 ± 0.4 [a] | 0.24 ± 0.034 [c] | 0.25 ± 0.031 [c] | 6.11 ± 0.52 [a] | 6.09 ± 0.70 [a] |

| Treatment | Ca (%) | | Na (%) | | Zn (ppm) | | B (ppm) | |
|---|---|---|---|---|---|---|---|---|
| | S1 | S2 | S1 | S1 | S2 | S1 | S1 | S2 |
| T0 | 1.42 ± 0.19 [d] | 1.54 ± 0.22 [e] | 3.42 ± 0.57 [d] | 3.81 ± 0.38 [d] | 26.4 ± 5.1 [e] | 25.1 ± 2.7 [f] | 26.8 ± 3.1 [d] | 28.2 ± 6.9 [d] |
| T1 | 2.58 ± 0.38 [b] | 2.81 ± 0.25 [b] | 6.09 ± 0.42 [a] | 6.07 ± 0.36 [a] | 74.5 ± 9.9 [c] | 77.3 ± 5.27 [c] | 45.4 ± 5.22 [a] | 42.3 ± 4.30 [b] |
| T2 | 2.57 ± 0.29 [b] | 2.21 ± 0.23 [d] | 5.46 ± 0.62 [b] | 5.51 ± 0.59 [b] | 85.2 ± 9.7 [b] | 82.7 ± 9.5b | 41.8 ± 4.09 [b] | 43.5 ± 3.44 [b] |
| T3 | 2.59 ± 0.20 [b] | 2.77 ± 0.49 [c] | 5.50 ± 0.48 [b] | 5.49 ± 0.62 [b] | 59.3 ± 6.9 [d] | 56.6 ± 6.10 [e] | 37.5 ± 3.29 [c] | 40.2 ± 2.1 [b] |
| T4 | 2.54 ± 0.35 [c] | 2.71 ± 0.30 [c] | 4.78 ± 0.55 [c] | 4.81 ± 0.68 [c] | 72.5 ± 7.8 [c] | 70.4 ± 12.4 [c] | 47.2 ± 6.46 [a] | 45.6 ± 5.22 [a] |
| T5 | 2.64 ± 0.52 [b] | 2.89 ± 0.66 [b] | 6.11 ± 0.71 [a] | 6.19 ± 0.68 [a] | 83.4 ± 4.1 [b] | 81.6 ± 9.41 [b] | 47.6 ± 4.5 [a] | 46.7 ± 3.65 [a] |
| T6 | 2.85 ± 0.43 [a] | 3.01 ± 0.57 [a] | 6.54 ± 0.80 [a] | 6.38 ± 0.81 [a] | 102.1 ± 15.4 [a] | 99.3 ± 14.1a | 49.3 ± 6.1 [a] | 48.2 ± 5.41 [a] |

Values with different letters show significant differences at $p \leq 0.05$ (LSD). (T0) = control, (T1) = n-Zeolite, (T2) = n-Zn, (T3) = n-B, (T4) = n-Si, (T5) = n-Zn + Si +B, (T6) = n-zeolite + Zn + Si + B., (n) means nanoparticles, (S) = seasons '1 and 2'.

Maximum plant tissue concentration of N, P, and B were recorded in T6 and T5 as compared to other treatments. Furthermore, the highest Ca and Zn concentrations in plant tissue were observed in plants treated with T6 at both seasons under salt stress, as compared to other treatments (Table 5). The increase of the endogenous nutrients' content of treated potato plants with nanoparticles could be associated with different mechanisms such as: (1) improving the root length and size, which increases nutrient uptake [21,65], (2) enhancing N metabolism such as increasing nitrate levels, reducing the activity of nitrate reductase activity, and N fixation of plant and microorganisms [90] (3) reducing the soil absorption of some nutrients especially P, particularly at low pH, and, thus, increasing the plant-available portion of some nutrients (P) in the soil, and controlling stomatal movement and transpiration of leaves improving mobility of soil nutrients towards the roots [91].

Silicon application was reported to stimulate root activity and plant vigor through increasing nitrogen uptake in barley plants subjected to salinity [50]. Moreover, nano-ZnO $Fe_3O_4$ application to *Moringa peregrina* under salinity stress was reported to increase N, P, $K^+$, $Ca^{2+}$, $Mg^{2+}$, Fe, and Zn contents and decrease $Na^+$ and $Cl^-$ contents [80]. In fact, Saffaryazdi et al. [92] postulated that Si could increase nitrogen contents by controlling chlorine uptake. Regarding leaf P content, Liang et al. [50] documented that total P content in barley increased by application of Si under salinity stress. The authors hypothesized that this increase may be related to the stimulated root activity as reflected by root dehydrogenase activity. Also, P bioavailability might be improved by Si because of chemical competition for the absorption sites between dihydrogen phosphate and silicate anions [50]. Likewise, studies have documented that nano-Zn seems to increase P uptake. Nano-ZnO particles were reported to enhance the phosphatase enzyme activity, as well as phytase, in mung bean [58]. In these two phosphorus solubilizing enzymes, Zn acts as a cofactor, thus contributing to the higher P uptake. Regarding boron and salinity interaction, it was reported that high contents of $Na^+$ and B in soil reduce the nutritional imbalance caused by the presence of each factor alone [87]. Concerning the effect of zeolite, it is well-established that, when applied to normal soil, it enhances soil nutrient retention with no alteration of drainage [59]. On the other hand, under salinity conditions, amendment with zeolite was revealed to increase both macro and trace element content ($Ca^{2+}$, $Fe^{2+}$, and $Mn^{2+}$) of salt-stressed barley [60]. Also, our results show that zeolite significantly increases $Ca^{2+}$ leaf concentration, apparently due to an increased availability of $Ca^{2+}$ in the soil. This leads to a decrease in the $Na^+/Ca^{2+}$ ratio in plant tissues and higher tolerance to $Na^+$ cations. The decreased $Na^+/Ca^{2+}$ ratio is an important salinity adaptation mechanism because the second stage of plant response to salt stress is ion toxicity, which starts when $Na^+$ cations accumulate at toxic concentrations in the leaves [49]. The ability of plants to reduce $Na^+$ uptake (influx) or increase $Na^+$ exclusion (efflux) is widely considered to be a main trait of variability in salinity tolerance within different species [49]. On the other hand, another important factor determining salinity tolerance is the ability of plants to sustain a high $K^+/Na^+$ ratio [93]. Interestingly, our results showed that the sodium ($Na^+$) content increased in the nanoparticles-treated plants as compared to control. This increase in $Na^+$ content in plant tissue may be possibly explained by an increase in the expression of genes that encode for V-$H^+$-ATPase and the V-$H^+$-PPase cells [94]. Although not measured in the present work, the increase in these two proton pumps might have energized the $Na^+(K^+)/H^+$ antiport activity at the tonoplast of vacuoles present in potato cells, which promoted the sequestration of high concentration of $Na^+$ in vacuoles through the action of NHX transporters [95,96].

### 3.4. Impacts of Soil Salinity on Leaf Proline, Gibberellic Acid and Abscisic Acid Contents

As shown in Table 6, proline, abscisic acid (ABA), and gibberellic acid (GA3) in potato leaves were also significantly affected by soil salinity and treatments ($p > 0.05$). In both seasons, all nanoparticle treatments improved the proline content. The highest proline content was found in T5 treatment while the lowest was recorded in the control (T0).

**Table 6.** Leaf proline and hormone concentration of potato plants grown in saline soil due to nanoparticles application.

| Treatment | Proline Content (mg g$^{-1}$) Leaves | | GA3 (µg g$^{-1}$ F W/Leaves) | | ABA (µg g$^{-1}$ F W Leaves) | |
|---|---|---|---|---|---|---|
| | S$^1$ | S$^2$ | S$^1$ | S$^2$ | S$^1$ | S$^2$ |
| T0 | 4.12 ± 0.81 [e] | 3.74 ± 0.4 [e] | 6.65 ± 0.71 [e] | 6.92 ± 0.58 [e] | 5.881 ± 0.74 [a] | 7.982 ± 0.91 [a] |
| T1 | 6.44 ± 1.04 [b] | 6.42 ± 1.8 [b] | 10.32 ± 2.07 [c] | 9.92 ± 1.95 [b] | 4.221 ± 0.82 [b] | 4.131 ± 0.76 [c] |
| T2 | 5.03 ± 0.095 [d] | 4.84 ± 0.99 [d] | 10.41 ± 1.89 [c] | 10.37 ± 2.21 [b] | 4.232 ± 0.77 [b] | 4.201 ± 0.91 [c] |
| T3 | 5.37 ± 0.85 [c] | 5.28 ± 1.01 [c] | 8.97 ± ± 1.71 [d] | 9.12 ± 2.01 [c] | 4.155 ± 0.83 [b] | 4.193 ± 0.53 [c] |
| T4 | 5.09 ± 0.93 [d] | 5.1 ± 0.98 [c] | 8.93 ± 1.80 [d] | 9.07 ± 1.75 [c] | 4.311 ± 0.79 [b] | 4.352 ± 0.98 [c] |
| T5 | 7.24 ± 1.31 [a] | 7.53 ± 1.17 [a] | 12.68 ± 2.28 [b] | 13.22 ± 2.80 [a] | 3.115 ± 0.69 [c] | 2.742 ± 0.36 [d] |
| T6 | 6.52 ± 1.16 [b] | 6.33 ± 1.13 [b] | 13.53 ± 3.43 [a] | 13.62 ± 4.01 [a] | 2.053 ± 0.38 [d] | 2.016 ± 0.29 [d] |

Values with different letters show significant differences at $p \leq 0.05$ (LSD). (T0) = control, (T1) = n-Zeolite, (T2) = n-Zn, (T3) = n-B, (T4) = n-Si, (T5) = n-Zn + Si +B, (T6) = n-zeolite + Zn + Si + B, (n) = nanoparticles, (S) = seasons '1 and 2'.

Plants synthesize and accumulate several organic osmolytes of low-molecular-weight such as proline in response to stressful environments, [49]. These molecules are called compatible solutes because they do not impede any regular metabolic process in plant cells. They also play an important role in alleviating the toxic effects of high Na$^+$ concentrations on enzymes, proteins or membranes under salinity. These may also play a role as oxygen radical scavengers to reduce the detrimental effect of ROS under high salinity [49]. Our results clearly showed that proline accumulation increases greatly with nanoparticles applications. Hussein et al. [27] affirmed that addition of nano-SiO$_2$ to salt stressed tomato improved leaf proline accumulation. Similar results in squash were reported by [28] as well. Moreover, the use of ZnO.Fe$_3$O$_4$ was reported to increase proline contents in *Moringa peregrina* under salinity [81].

Similarly, such tendency was noted in GA3 content, where the maximum values of GA3 were recorded in plants treated with combined nanoparticles (T5 and T6) while the minimum values were found in control plants in the two seasons (Table 6), during two years of experimentation. On the other hand, the greatest accumulations of abscisic acid (ABA) in potato leaves were observed in control plants (T0) while the least values were recorded in combined treatments (T5 and T6). Our results showed that application of combined nanoparticle treatments to salt-stressed potato enhanced the concentration of endogenous GA3 and lowered the content of ABA. It was reported that Si has an effect on plant hormones under salinity [97,98]. In hydroponic salt-stressed soybean plants, [97] cited that ABA content decreased when Si was applied. Conversely, the authors observed a decrease in gibberellin levels of salt-stressed soybean which significantly increased with the addition of Si. Nevertheless, the authors revealed that after 6 and 12 h, the genes related to ABA biosynthesis (*OsZEP* and *OsNCED1*) were induced by Si treatment under salt stress. This implies that Si partly promotes salinity tolerance in plants through regulation of hormone gene expression under salinity.

*3.5. Impacts of Soil Salinity on Number and Yield of Potato Tubers*

Number of potato tubers per plant along with tuber weight is a crucial index of the total yield of the potato crop. The data concerning the number of potato tubers per plant is presented in (Figure 3).

A higher number of potato tubers per plant was found in all treatments, as compared to control. Compared to the other treatments, the maximum number of potato tubers was found in the combined treatments (T5 and T6, Figure 3A). In the first season, the numbers of potato tubers per plant in combined treatments (T5 and T6) were 6.1 and 6.7, respectively, followed by T1 (= 5.50), T2 (= 4.5), T3 (= 4.4) and T4 (= 4.5) as compared to control (T0 = 3.3). A similar trend was also noted in the second season for the values of tuber number in T1, T2, T3, and T4, as compared to control. Similar results were noted in tuber yield (Figure 3B). The maximum tuber yield was found in T5 and T6 while the minimum was observed in the control in both seasons. The results clearly showed that soil salinity negatively affected the tuber yield of the untreated plants more than the treated ones. It was observed that the yield reduction was associated with the reduction of the number, size, and weight of tuber per plant [99].

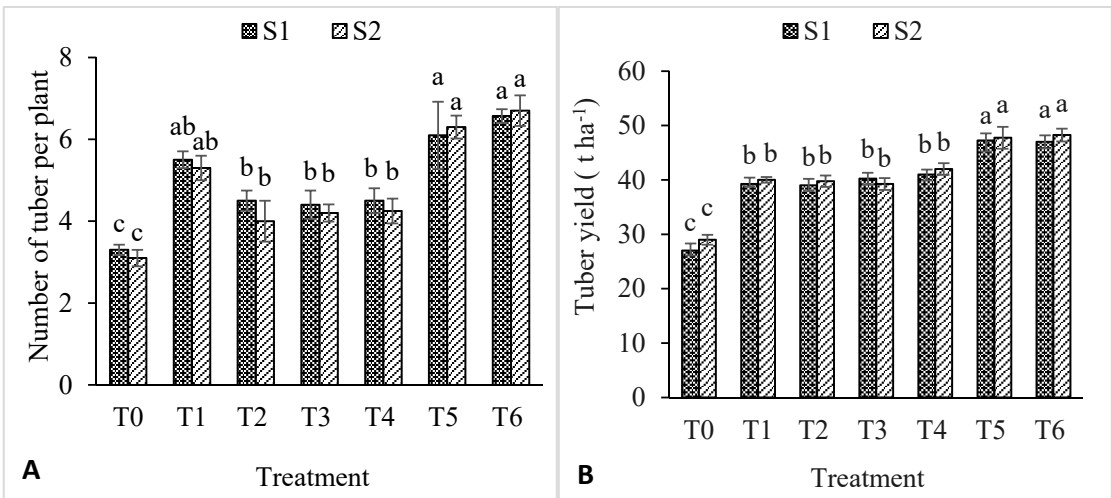

**Figure 3.** (**A**) tuber number and (**B**) tuber yield in saline soil due to nanoparticles application. Columns with similar letters show insignificant differences at $p \leq 0.05$ (LSD). Bar indicates to standard deviation (± SD). Treatments were organized as follow: (T0) = control, (T1) = n-Zeolite, (T2) = n-Zn, (T3) = n-B, (T4) = n-Si, (T5) = n-Zn + Si +B, (T6) = n-zeolite + Zn + Si + B., (S) = seasons '1 and 2'.

These results are in accordance with Nagaz et al. [100] who concluded that lower tuber yields under salinity are due to the accumulation of $Na^+$ and $Cl^-$ in the plant cell. On the contrary, supplementary application of Zn, Si, B, and zeolite nanoparticles apparently improve the morphological and physiological attributes and alleviate salt stress [67,101]. In potato, long-term field trials showed that the addition of $K_2O_3Si$ increased the potato yield by 12.3% [102]. In the case of zinc, application of 1500 ppm of nano-ZnO on corn plants enhanced the yield by 42% as compared to control plants [103]. In addition, the yield of field-grown cucumber plants was increased by 36% using 5 mg kg$^{-1}$ nano-ZnO [104]. Regarding boron, an antagonistic effect between salinity and boron was reported to reduce the toxicity of NaCl in pepper plants [86]. Also, combined application of zeolite, Si, and selenium was revealed to improve yield and yield components of salt-stressed canola plants such as harvest index and oil percentage [89].

### 3.6. Impacts of Soil Salinity on Chemical Compositions of Potato Tubers

As shown in Table 7, the chemical composition of tubers was significantly affected by soil salinity and supplementary application of Zn, Si, B, and Zeolite nanoparticles. Under soil salinity stress, the highest carbohydrate content in tubers was recorded in the combined treatments (T5 and T6). There was no significant difference among single treatments (T1, T2, T3, and T4) as compared to control (T0) during both seasons. In contrast, the maximum starch content in tuber was found in the untreated plants (T0) while the minimum was observed in the combined treatment (T5 and T6).

Tuber protein content significantly increased with application of Zn, Si, B, and Zeolite nanoparticles ($p \geq 0.05$). The highest values of protein in tubers were recorded in T5 and T6 while the lowest values were found in the untreated plants (T0). The peroxidase (POD) and polyphenol oxidase (PPO) exhibited similar trend. Compared to the control (T0), the maximum POD and PPO activities in potato tuber were recorded in T5 and T6 treatments followed by the single treatments application (T1, T2, T3, and T4), during the two seasons. Improvement of protein and carbohydrate content in potato yield of treated plants might be due to the contribution of Zn, Si, and B nanoparticles in the activation of several enzymes associated with protein biosynthesis and carbohydrate metabolism [105]. Also, zeolite alone increased the total soluble sugars of canola [89], and when it was combined with Si, it increased the total solids content of onion. In addition, the nanoparticles play an important role in enhancing the leaf photosynthesis rate and $CO_2$ assimilation, as mentioned before, which increases the accumulation of carbohydrate and protein content in the economic parts [73–75].

**Table 7.** Tuber chemical composition of potato plants grown in saline soil due to nanoparticles application.

| Treatment | Starch Content (%) | | Carbohydrates (%) | | Protein Content (%) | | POD (Units mg$^{-1}$Protein) | | PPO (Units mg$^{-1}$ Protein) | |
|---|---|---|---|---|---|---|---|---|---|---|
| | S$^1$ | S$^2$ | S$^1$ | S$^2$ | S$^1$ | S$^2$ | S$^1$ | S$^2$ | S$^1$ | S$^2$ |
| T0 | 80.4 ± 10.3 [a] | 81.6 ± 11.3 [a] | 72.14 ± 6.8 [b] | 71.58 ± 4.8 [b] | 0.83 ± 0.08 [d] | 0.72 ± 0.07 [e] | 0.010 ± 0.0005 [e] | 0.008 ± 0.0001 [f] | 6.81 ± 0.92 [d] | 7.01 ± 0.52 [d] |
| T1 | 67.8 ± 7.4 [c] | 61.5 ± 6.9 [c] | 75.22 ± 5.7 [b] | 74.48 ± 5.2 [b] | 2.20 ± 0.47 [b] | 2.79 ± 0.61 [b] | 0.037 ± 0.0018 [b] | 0.035 ± 0.0006 [c] | 7.28 ± 0.72 [b] | 7.27 ± 0.87 [b] |
| T2 | 70.3 ± 9.2 [b] | 70.1 ± 8.5 [b] | 74.66 ± 7.2 [b] | 75.06 ± 6.6 [b] | 2.04 ± 0.21 [b] | 2.045 ± 0.30 [c] | 0.032 ± 0.0009 [b] | 0.031 ± 0.0003 [c] | 7.19 ± 0.83 [b] | 7.22 ± 0.80 [b] |
| T3 | 68.5 ± 5.1 [c] | 68.3 ± 7.6 [b] | 72.69 ± 6.5 [b] | 73.15 ± 6.9 [b] | 2.1 ± 0.56 [b] | 2.21 ± 0.37 [b] | 0.040 ± 0.0005 [a] | 0.042 ± 0.0081 [b] | 7.15 ± 1.01 [c] | 7.19 ± 1.4 [c] |
| T4 | 69.4 ± 8.7 [c] | 68.7 ± 6.39 [b] | 73.91 ± 8.4 [b] | 73.96 ± 5.6 [b] | 1.91 ± 0.19 [c] | 1.75 ± 0.14 [d] | 0.027 ± 0.0015 [c] | 0.025 ± 0.0001 [d] | 7.14 ± 0.99 [c] | 7.18 ± 0.57 [c] |
| T5 | 58.7 ± 6.69 [d] | 54.9 ± 7.3 [d] | 78.51 ± 7.5 [a] | 77.89 ± 6.3 [a] | 2.58 ± 0.15 [a] | 2.64 ± 0.51 [a] | 0.040 ± 0.0026 [a] | 0.048 ± 0.0009 [a] | 7.52 ± 1.01 [a] | 7.45 ± 1.8 [a] |
| T6 | 56.3 ± 5.9 [d] | 53.2 ± 4.9 [d] | 81.36 ± 9.1 [a] | 85.72 ± 7.7 [a] | 3.21 ± 0.54 [a] | 2.89 ± 0.88 [a] | 0.044 ± 0.00088 [a] | 0.051 ± 0.0006 [a] | 7.48 ± 1.3 [a] | 7.50 ± 2.01 [a] |

Values with different letters show significant differences at $p \leq 0.05$ (LSD). (T0) = control, (T1) = n-Zeolite, (T2) = n-Zn, (T3) = n-B, (T4) = n-Si, (T5) = n-Zn + Si +B, (T6) = n-zeolite + Zn + Si + B, (n) = nanoparticles, (S) = seasons '1 and 2', (POD) = Peroxidase, (PPO) Polyphenoloxidase.

Salt toxicity leads to oxidative damage and membrane lipid peroxidation [71,105,106]. Plasma membrane injury is associated with an increased production of reactive oxygen species ROS [46]. Removing (ROS) activity mainly occurs by antioxidant enzymes such as PPO, POD, catalase (CAT), and ascorbate peroxidase (APX) [107]. Liang et al. [50] Revealed that Si could induce the activity of superoxide dismutase enzyme and lower malondialdehyde contents in barley plants under salinity. This leads to lower oxidative damage to membrane lipids. This is probably due to an increased production of this clearly demonstrating that Si is implicated in promoting different systems of antioxidation. However, the exact mechanism mediated by Si under salt stress needs further investigation at both genetic and transcriptional levels. Also, nano-$SiO_2$ was revealed to increase the activity of antioxidant enzymes which improved the plant tolerance to salinity stress [27,28]. Moreover, combined zeolite and Si increase the activities of several important antioxidant enzymes [108]. As to zinc, when wheat plants were treated with 500 ppm of nano zinc oxide, a marked increase in peroxidase activity and lignification of root cells was found [109]. Similarly, an increased superoxide dismutase (SOD) and POD enzymes activities using nano-ZnO (25–200 mg $L^{-1}$) in cotton plants, as compared to control plants, was reported [110].

All in all, the combined effects of Zn, Si, B, and zeolite nanoparticles used in the present study lead ultimately to a higher yield of potato tubers, the economic parts of the plant. According to our study, this improved yield can be attributed so far to several factors such as: (a) improved activity of photosynthesis, (b) enhanced ratio of $Na^+/Ca^{2+}$, (c) increased activity of antioxidant enzymes, (d) increased soluble substances concentration in plant tissues, resulting in better sodium tolerance in plants, and (f) improved carbohydrate metabolism and transportation [21,105,111–114].

## 4. Conclusions

The results of the present research provide evidence demonstrating the beneficial effects of applying Si, Zn, B, and zeolite nano-particles, single or combined, on alleviating the negative effects of soil salinity on potato plant growth, physiology, and tuber yield. A remarkable synergistic effect was observed when a combination of zeolite, Si, Zn, and B nanoparticles was applied to salt-stressed potato plants. Acting together, these nanoparticles significantly boosted retention of water and nutrients and induced enzymatic antioxidant activities in salt-stressed potato plants. These effects promoted a higher tolerance to salinity in potato plants which consequently, boosted growth and tuber yield by increasing nutrient use efficiency and photosynthetic activity. The scientific basis of how each of our treatments may have promoted the plant tolerance to salinity was discussed in this work. Nevertheless, the synergistic effect of the combined treatments still needs further investigation. Such an effective approach is therefore presented as a possible solution to face the increasing soil salinity problem throughout the world. Further long-term studies on potato and other strategic crops are recommended to show the relationship between nano-particle dose, soil type, and ecological impact.

**Author Contributions:** A.W.M.M. and S.A.M. conceived the experiments and methodology; A.W.M.M., S.A.M., E.A.A., S.M.A. and M.B.I.E.-S. performed the experiments, measurements and analyses; A.W.M.M., E.A.A. and S.A.M. curated and analyzed the data; A.W.M.M., E.A.A. and S.A.M. wrote the manuscript draft; E.A.A., M.B.I.E.-S. and S.M.A. reviewed, edited and completed the manuscript; A.W.M.M. supervised the whole work. All authors have read and agreed to the published version of the manuscript.

**Funding:** The present work was funded by Cairo University, Egypt, "Application of nanotechnology in saline soil reclamation with strategic crops" project.

**Acknowledgments:** We would like to show our appreciation and express our deepest gratitude towards Cairo University and Faculty of Agriculture, Research Park (FA-CURP), for sample analyses and facilities.

**Conflicts of Interest:** The authors declare no conflict of interest.

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
