# Peer review of "Synergetic Effects of Zinc, Boron, Silicon, and Zeolite Nanoparticles on Confer Tolerance in Potato Plants Subjected to Salinity"

_agronomy, doi:10.3390/agronomy10010019_

Round 1

Reviewer 1 Report

Referee’s Report

Manuscript ID: agronomy-646472

“Synergetic Effects of Zinc, Boron, Silicon and Zeolite Nanoparticles on Confer Tolerance in Potato Plants Subjected to Salinity”.

The paper covers a very interesting topic, that is the use of innovative methods such as nanoparticles application to vegetables crops, in order to alleviate and minimize salinity stress. However, I am not convinced that this study is suitable for a publication in the present version.

The paper often appears unclear so that a deep revision of the English language in all the sections should be carried out, in order to enhance the text readability and its comprehension.

I believe that the text would benefit a more clear and well organized report of the study carried out.

Moreover, the paper presents some limitations such as:

Introduction is weak with particular reference to the nanoparticles-salinity interactions, as also to previous findings of nanoparticles application to the variables tested in this experiment and under similar experimental conditions.

I will prefer to read more on nanoparticles and its effects on salinity stress control.

Materials and Methods the experimental layout should be better described; a more detailed description of the adopted experimental treatments should be reported; a more detailed description of the collected experimental data should be reported (when and how they were collected); the measure unit tons/fed should be revised, expressing ita s tons/ha; it should be specified what is the compost added to the soil prior to potato planting; more information about soil salinity should be added; more information about irrigation water characteristics should be reported font size of tables 1, 2, 3; of tables 1, 2, 3 heading; of figures 1, 2, 3 caption should be reported according to the instruction for the authors provided by the Journal in the template file.

All these aspects should be better organized and presented.

Results and Discussion It is often difficult to follow what is reported as well as the meaning and the extent of the obtained results, so the discussion would benefit considerations of the mechanisms responsible for the author's observations. font size of tables, table headings; figure captions should be revised and reported according to the instruction for the authors provided by the Journal in the template file

Conclusions a sentence with useful informations and perspectives to the readers should be added. References font size should be revised and reported according to the instruction for the authors provided by the Journal in the template file the hypen between two author’s names should be deleted

Author Response

Dear Prof.Doctor:

Thank you very much for the valuable reviewers comments which helped us to significantly improve the quality of our manuscript.

Kindly find below some general answers to the comments of the three respectful reviewers.

We have done the following modifications to the manuscript:

Complete English language editing of the manuscript by a professional editor that amended all the grammatical, style and flow mistakes of the manuscript. The unclear parts of the manuscript were completely modified to provide a consistent and smooth flow of information. Clarifying the Materials and Methods section and adding relevant information on the methodology of work. A substantial number of citations have been added to the Introduction and Results & Discussion sections in order to enrich the manuscript integrity. Additional details were added to conclusion section to summarize better the work.

All the above mentioned modifications have been inserted into the manuscript in red color.

We would also like clarify that an alteration of manuscript formatting occurred automatically after the manuscript was uploaded to the website, which changed manuscript font type and size, as well as table and figures formatting. As you see, that was out of our control, and in all cases we have corrected all these issues.

Thank you very much once again, and we hope that the changes made to the manuscript will meet your expectations.

Sincerely yours,

Reviewer 2 Report

While the research described in this manuscript proves to be interesting and can contribute to the field of study, it is not ready to be published in its current state. The following suggestions are general and do not encompass every correction that should be made. Significant revisions and further review are needed. 

Overall:

English and grammar need to be improved throughout the document. Formatting needs to be consistent and significantly improved. Many discrepancies in font size, graph placement, etc. need to be addressed. SI units should also be used throughout.

Introduction:

Formatting and english/grammar need to be addressed but most are minor. Improve clarity in lines 65-68, the message is not clear.

Materials and methods:

Significant changes need to be made in this section.  Among these include improving english and grammar, sentence structure, and flow. 

The format is very choppy and sections should be combined. The methods were generally unclear and it was very difficult to determine how the experiment was conducted.

More details were needed in many of the sections and references were also lacking.

The authors also need to use SI units for all applicable measurements.

The tables were larger font that the rest of the text and should be formatted to fit on one page if possible. About half way through the materials and methods, the font size changed and needs to be addressed. 

Results and discussion:

The results were not fully summarized and lacked many of the details that would really contribute to the message of the research. The results and discussion should be thoroughly revised to improve this manuscript.

The table formatting made it very difficult to interpret the results. Figures should have the X axis labeled and include units on the Y axis.  Treatments should be labeled on the figures as well. There are floating boxes in the figures and this is not explained and it interferes with interpretation. 

Add statistics to figures or captions (pvalues) or text where relevant. 

Improve english and grammar throughout.  

Work on the flow and data interpretation.

Author Response

Dear Professor Doctor:

Thank you very much for the valuable reviewers comments which helped us to significantly improve the quality of our manuscript.

Kindly find below some general answers to the comments of the three respectful reviewers.

We have done the following modifications to the manuscript:

Complete English language editing of the manuscript by a professional editor that amended all the grammatical, style and flow mistakes of the manuscript. The unclear parts of the manuscript were completely modified to provide a consistent and smooth flow of information. Clarifying the Materials and Methods section and adding relevant information on the methodology of work. A substantial number of citations have been added to the Introduction and Results & Discussion sections in order to enrich the manuscript integrity. Additional details were added to conclusion section to summarize better the work.

All the above mentioned modifications have been inserted into the manuscript in red color.

We would also like clarify that an alteration of manuscript formatting occurred automatically after the manuscript was uploaded to the website, which changed manuscript font type and size, as well as table and figures formatting. As you see, that was out of our control, and in all cases we have corrected all these issues.

Thank you very much once again, and we hope that the changes made to the manuscript will meet your expectations.

Sincerely yours,

Reviewer 3 Report

The manuscript titled “Synergetic Effects of Zinc, Boron, Silicon and Zeolite Nanoparticles on Confer Tolerance in Potato Plants Subjected to Salinity” to be published in Agronomy aims to investigate and highlight the protective effects exerted by nanoparticles on Solanum tuberosum cv Diamont plants exposed to salt stress. The protection is exerted by the application of specific nanoparticles to the soil and the authors measured various morphological traits and physiological, biochemical and photosynthetic aspects. Since this abiotic stress is a serious and emerging problem for cultivated plants, strategies adopted to protect plants against this stress or to increase the tolerance of crops toward salt stress are extremely relevant and important. Globally, the research topic is very interesting and it provides many results and evidences to the scientific community. However, the manuscript present many grammar mistakes, many sentences are little clear and the images, tables and figures are not good for a publication. For example: images lack of the captions and figure 1 does not show any axes and in turn any values. For these reasons, the manuscript is accepted with major revisions, because the amount of data is high, but the manuscript should be better written.

Therefore, the authors should:

deeply revise the English language by sending the manuscript to an English-speaking native or to a service; correct all tables and figures making them available for publications.

Then, I would like to revise the manuscript once again.

Author Response

(The authors gave the same response as above.)

Round 2

Reviewer 1 Report

Referee’s Report

Manuscript ID: agronomy-646472

“Synergetic Effects of Zinc, Boron, Silicon and Zeolite Nanoparticles on Confer Tolerance in Potato Plants Subjected to Salinity”.

The quality of the manuscript has been improved. I recommend to accept this paper for publication after making minor changes as below indicated.

Abstarct

Line 28: eliminate “As”

Line 32: replace “mentioned” with “aferomentioned”

Introduction

Line 39: eliminate “of land”

Line 66: replace “Nanoparticles” with “nanoparticles”

Line 75: eliminate “all”

Line 79: replace “widely- reported” with “widely reported” ans “salt- stressed” with “salt stressed”

Line 84: report author name before reference number

Line 88: report author name before reference number

Materials and Methods

Line 98: replace “newly- reclaimed” with “newly reclaimed

Line 101: replace “mechanical” with “physical”

Line 203: specify what type of compost you have applied

Line 107: replace “done” with “carried out”

Line 109: replace “done” with “carried out”

Table 1: report all the measure units in brackets; report “available nutrients” in bold

Table 2: replace “PH” with “pH”; report all the measure units in brackets; replace dsm-1 with “dS m-1”; replace “cfu” with “CFU”; check if Total N measure uniti is % or ‰

Lines 149 and 150: replace “ionized water” with “deionized water”

Lines 150 and 151: correct “3hours” as “3 hours”

Line 156: correct “cm3” as “cm3

Line 157: replace “milled podwers” with “podwers milled”

Line 213: correct “Co2” as “CO2

Line 220: replace “70°C” with “70 °C”

Line 222: replace “herbs” with “samples”

Line 223: eliminate “to the samples”

Line 230: replace “herbs” with “samples”

Line 232: replace “watmann” with “Whatman”

Line 232: replace “HCL” with “HCl”

Line 232: replace “0.5 ml of aliquot sample” with “0.5 ml sample aliquot”

Line 246: replace “10ml” with “10 ml”

Line 256: replace “herbs” with “samples”

Line 269: full stop after [45]

Line 270: replace “mixture is” with “mixture was”

Results and Discussion

Line 321: replace “in armony ” with “in accordance”

Line 323: what does mean “In the first phase of salinity”; revise it, otherwise delete it

Line 335: report the authors name before [53]

Line 337: report the authors name before [54]

Line 342: report the authors name before [48]

Line 346: report the authors name before [55]

Line 348: report the authors name before [56]

Line 363: report the authors name before [59]

Table 4 heading: replace “Changes in vegetative growth” with “Growth parameters”

Line 363: replace “present results” with “experimental results”

Line 380: replace “affected soil” with “affected by soil”

Line 382: delete the comma after (2A)

Line 386: report the authors name before [64]

Line 395: report the authors name before [70]

Line 397: replace “revealed” with £reported”

Line 399: report the authors name before [72]

Line 408: add “conditions” after “salinity”

Line 408: delete full stop in “ZnO.Fe3O4

Line 414: delete “Changes in”

Line 416: replace “insignificant” with “not significant”

Line 417: replace “indicates to standard” with “indicates standard”

Line 421: report the authors name before [76]

Line 422: what is Fv/Fm?

Line 454: delete full stop in “ZnO.Fe3O4

Line 456: report the authors name before [86]

Line 456: replace “Chlorine” with “chlorine”

Line 457: report the authors name before [53]

Line 461: replace “reports” with “studies

Line 486. Delete “Changes in”

Line 494: replace “Leaves” with “leaves”

Line 545: replace “In contrary” with “On the contrary”

Line 587: what are CAT and APX?

Line 587: what is SOD?

Line 610. Delete “Changes in”

Conclusions

Line 616: delete “paper”; delete “an”

Line 619: replace “revealed” with “observed”

Line 626: replace “is still in need” with “still nedd”

Line 627: replace “a cost-effective” with “effective”

Lines 630-631: delete “these nanoparticles in various crops”

Lines 632: report author contributions; It lacks!

Author Response

Dear respectable Prof.Doctor

Thank you very much for the comments (second round) of revision of our manuscript. The meticulous revision of the respected reviewers greatly helped us to amend the few remaining mistakes of the manuscript.

We have corrected the mistakes and fixed some numbering errors of some references. Please kindly note that all the new changes of the second round of comments were inserted in the manuscript in green color.

We hope this corrected version will be suitable for publication in your respected journal.

Sincerely yours,

Reviewer 2 Report

The manuscript has significantly improved. See attached suggestions and comments for minor grammatical and formatting improvements.

Author Response

(The authors gave the same response as above.)

Reviewer 3 Report

The manuscript titled “Synergetic Effects of Zinc, Boron, Silicon and Zeolite Nanoparticles on Confer Tolerance in Potato Plants Subjected to Salinity” has been deeply modified and now the manuscript is more complete. On the other hand, it still suffers of many grammar mistakes and I had some troubles to read and understand the meaning of many sentences and this is a pity because the authors made a lot of experiments and they showed many useful data for a very interesting and intriguing topic for the challenge against the salt stress. Therefore, the manuscript should be revised again by the authors to avoid and correct grammar mistakes.

Please modify the following sentence:

Along the whole manuscript, please uniform the reference format (bold? Because some references are written in bold while others in normal character); At row 28, please delete the bracket; At row 99,please delete the space between “( and >3”; At row 90, please delete “regulating” because it is repeated twice; At row 106: how many plants were analysed per year? How many biological replicated were made? For each experiment and for each figure and table: please specify the plant organ that was analysed (in many experiments this data is reported, but in others not). At row 152, please delete the space between “( and 7”; At row 153, please rewrite “B2O3”; At row 154, please delete the space between “32: and 1”; At row 156, please rewrite “cm3”; At row 159 and 162, rewrite “°C”; At rows 200, 213 and 218, please rewrite “CO2” At row 218, please modify “mol-1”; At row 223, please insert a space between “5 and ml” Form row 146 to 169, please correct some grammar errors (spaces, super- and subscript numbers, and so on) and insert precise values of the weighed powders. At row 224, please delete the dot. Form row 225 to 229, please correct the paragraph format (double line). At row 231, please correct “mg”; At row 234, please correct “HCl”; The sentence at row 242 is little clear; At row 258: the meaning of “(1x3 h. 2x1 h)” is little clear; At row 258, please delete the dot. At row 266, please delete the dot. At row 319, one measurement lacks (there are 3 values instead of 4); At row 339: is Basil the plant species reported by [54]? I found that the plant species was barley At row 366, please delete the space after the bracket; The sentences reported in rows 309-311 are not supported by any table or figure. The authors forgot to include a reference to the specific table (I guess it is the table 4). Table 4 should be better formatted: there is a space between the first and second horizontal line (the second line is over the various S) and the superscript numbers after “S” is without explanation in the caption. Moreover T2 is not vertically in line with the others “T”. At row 365, please change the term “indicate” with “support”; At row 382, please delete the comma; The vertical axes of the figure 2E lacks of the entire title At row 494, please rewrite “leaves” with lowercase; At row 510, please rewrite “affirmed” with lowercase; In the reference section, some references are bad wrote and they should be rewritten by using the right format (for example reference 8 and 10). Supplementary material is not cited in the main text and it does not have a caption.

Author Response

(The authors gave the same response as above.)
